# Mammary Gland Transcriptome and Proteome Modifications by Nutrient Restriction in Early Lactation Holstein Cows Challenged with Intra-Mammary Lipopolysaccharide

**DOI:** 10.3390/ijms20051156

**Published:** 2019-03-06

**Authors:** Karol Pawłowski, José A. A. Pires, Yannick Faulconnier, Christophe Chambon, Pierre Germon, Céline Boby, Christine Leroux

**Affiliations:** 1Université Clermont Auvergne, INRA, VetAgro Sup, UMR Herbivores, F-63122 Saint-Genès-Champanelle, France; karol_pawlowski@sggw.pl (K.P.); jose.pires@inra.fr (J.A.A.P.); yannick.faulconnier@inra.fr (Y.F.); celine.boby@inra.fr (C.B.); 2Department of Pathology and Veterinary Diagnostics, Faculty of Veterinary Medicine, Warsaw University of Life Sciences, 02-776 Warsaw, Poland; 3INRA, INRA, Plateforme d’Exploration du Métabolisme, composante protéomique PFEMcp), F-63122 Saint-Genès Champanelle, France; christophe.chambon@inra.fr; 4INRA Val de Loire, UMR ISP, F-37380 Nouzilly, France; pierre.germon@inra.fr; 5Department of Food Science and Technology, University of California Davis, Davis, CA 95616, USA

**Keywords:** nutrigenomics, transcriptomic analysis, proteomic analysis, mammary gland, cows, inflammation

## Abstract

The objective is to study the effects of nutrient restrictions, which induce a metabolic imbalance on the inflammatory response of the mammary gland in early lactation cows. The aim is to decipher the molecular mechanisms involved, by comparing a control, with a restriction group, a transcriptome and proteome, after an intra-mammary lipopolysaccharide challenge. Multi-parous cows were either allowed *ad libitum* intake of a lactation diet (*n* = 8), or a ration containing low nutrient density (*n* = 8; 48% barley straw and dry matter basis) for four days starting at 24 ± 3 days in milk. Three days after the initiation of their treatments, one healthy rear mammary quarter of 12 lactating cows was challenged with 50 µg of lipopolysaccharide (LPS). Transcriptomic and proteomic analyses were performed on mammary biopsies obtained 24 h after the LPS challenge, using bovine 44K microarrays, and nano-LC-MS/MS, respectively. Restriction-induced deficits in energy, led to a marked negative energy balance (41 versus 97 ± 15% of Net Energy for Lactation (NEL) requirements) and metabolic imbalance. A microarray analyses identified 25 differentially expressed genes in response to restriction, suggesting that restriction had modified mammary metabolism, specifically β-oxidation process. Proteomic analyses identified 53 differentially expressed proteins, which suggests that the modification of protein synthesis from mRNA splicing to folding. Under-nutrition influenced mammary gland expression of the genes involved in metabolism, thereby increasing β-oxidation and altering protein synthesis, which may affect the response to inflammation.

## 1. Introduction

Milk is synthesized in mammary glands (MG) involving a large number of genes, the expression of which is modulated at a nutritional level [1] and by the health status [2]. Mastitis is the inflammatory response of the mammary gland to pathogens. This pathology is one of the most prevalent disease and has considerable economic impact due to decreased milk production, discarded milk, cost of veterinary services, and culling [3]. Mastitis is caused by various microorganisms (bacteria, fungus, and viruses). Gram-positive bacteria, such as *Staphylococcus aureus* and *Streptococcus uberis* might cause persistent infections, with pathogens surviving inside host cells [4,5], whereas Gram-negative coliform bacteria, such as *Escherichia coli* most frequently cause an acute inflammation and, eventually, severe mastitis with clinical signs [6]. Cows are particularly susceptible to *E. coli* MG inflammation during the periparturient period, due to altered immune function [7]. Early lactation is often associated with metabolic disorders related to stress, energy deficit and mobilization of the body reserves, hypocalcemia, and metritis, which are likely to influence immune function [8,9,10]. Negative energy balance (NEB) affects the inflammatory response, which could be due to changes in the metabolic milieu, such as an increased concentration of circulating ketone bodies [8,11]. Undernutrition, however, had minor effects on the response to lipopolysaccharide (LPS) and *S. uberis* challenges in mid-lactation dairy cows [12,13]. The current study is conducted during early lactation Holstein cows, which is characterized by enhanced metabolic deviations in response to NEB that might influence immune system function.

Transcriptome and proteome profiling techniques are available to study inflammation-related changes and enhance the understanding of host–pathogen interactions. Previous research has employed reproducible protocols to challenge lactating cow udders with live pathogens, such as *E. coli*, *S. aureus,* or using LPS to induce an acute inflammatory response [12,13,14,15]. High-throughput gene expression technologies, as transcriptomic analyses, have been used to decipher the molecular mechanisms of MG response to inflammation [16], showing that LPS is a strong stimulator of gene expression in an inflammatory response [17]. Transcriptomic analyses showed that MG differentially expressed genes (DEG) 24 h after *E. coli* infection in early-lactating cows, showing the associations of up- and down-regulated genes, respectively with immune response functions and fat metabolism [18]. A study at mid-lactation showed a massive effect of *E. coli* infection on gene expression at 18 h post-infection in infected quarters, but also having an effect on gene expression in neighboring quarters [19]. Furthermore, hierarchic clustering of DEG, showed a sharp separation of the infected and the control group [19]. Similar results were observed in primary bovine mammary epithelial cells after challenging the *E. coli* or *S. aureus* [20,21,22]. The modification of bovine milk proteome were reported during inflammation, showing an increase in the concentration of proteins of blood serum origin as serum albumin is made up of antimicrobial peptides [23,24]. To the best of our knowledge, the effects of negative energy balance on the responses to acute inflammation have not been studied at a protein expression level in the MG of early lactation cows.

We hypothesized that aggravated undernutrition in early lactating cows modifies the inflammatory response at mRNA and protein levels. Thus, the objective of this study is to evaluate the effects of undernutrition, and the resulting metabolic imbalance on MG transcriptome and proteome in early lactation Holsteins, challenged with intra-mammary LPS. We used complementary approaches to study the effects of undernutrition with a first targeted study of genes involved in the inflammation response, using RT-qPCR, then two global analyses using transcriptomic and proteomic analyses. These two last approaches allowed us to investigate the holistic response of MG. The effects of LPS are already well-described in the literature, thus attention is focused on the effects of restriction during inflammation, by comparisons with MG transcriptome, and the proteome of control versus under-fed early lactating cows. This study was performed with the aim of increasing the knowledge-base on the effects of NEB occurring in early lactation which is the time of high risk of mastitis. A better understanding can help to design news strategies to prevent MG inflammation.

## 2. Results

### 2.1. Dietary and Inflammatory Challenges Influenced Milk and Blood Composition

Prior to diet treatments at 24 ± 3 days in milk (DIM), no differences were observed for intake, milk yield, composition and component yield, Net Energy for Lactation (NEL) balance, body score (BS), body weight (BW), plasma metabolite, and insulin concentrations. Feeding the ration containing 48% of straw (restricted group: REST) induced an immediate depression of dry matter intake (DMI) and decreased the energy balance from 5.2 ± 8.9 to −67.2 ± 18.9 MJ/day one day before (corresponding to day 23), and on the last day of restriction (day 27), respectively. The plasma concentrations of glucose and insulin decreased, whereas non-esterified fatty acids (NEFA) and beta-hydroxy butyrate (BHBA) increased dramatically in REST (Appendix A). The 96 h of nutrient restriction decreased milk yield from 37.9 to 22.4 kg/day (*p* < 0.001) and milk protein yield from 1.12 to 0.62 kg/day (*p* < 0.005) in REST. Milk fat percentage increased during feed restriction in REST from 4 to 5.5% (*p* < 0.001) and returned to pre-restriction concentrations on the same day of re-feeding a regular diet. In CONT cows, milk fat content was greatest only during the 48 h following LPS injection (*p* < 0.05) compared with all other DIM. These variables were unchanged in the control (CONT) cows [25]. Within 2 to 6 h after injection with lipopolysaccharide, we noticed the edema of the challenged quarter of all cows, an increase in rectal temperature up to 39.5 °C (temperature increment was +2.1 ± 0.15 °C). The effect of inflammation also was confirmed by milk somatic cell count (SCC). The day before the LPS challenge, whole udder composite milk (from PM and AM milking) SCC was 78 000/mL and 92 000 /mL (*p* = 0.51) in CONT, and in REST, respectively. The SCC response was greater in REST compared to CONT cows (6919 versus 1956 × 1000 per mL, respectively) in composite milk samples from the two milkings that followed LPS injection. SCC returned to pre-LPS counts, within less than 7 days post-LPS challenge and biopsy. Moreover, quarter milk IL-8, IL1-β, TNF-alpha, and CXCL3 at time zero (before LPS challenge) did not differ between CONT and REST, but their concentrations increased in response to LPS in both groups. This data shows that these indicators of mammary inflammation did not differ between CONT and REST before or after the challenge.

### 2.2. Effects of Under-nutrition on Expression of Genes Involved in Inflammation Response by RT-qPCR Analysis

RT-qPCR analysis was performed to quantify candidate genes (*CCL5*, *LAP*, *RBP4*, *IL8*, *IL1*, *STAT3*, *CD36*, and *TAP)* chosen on the basis of their implication in inflammation [26] Expression of these genes in mammary gland (MG) did not differ between REST and CONT (*p* ≥ 0.1) 24 h after the LPS challenge, except for the defensin Tracheal Antimicrobial Peptide (*TAP*) gene which tended to decrease in REST (*p* = 0.07). The expression of *INSIG1*, and *CSN2* genes, which are involved in the biosynthesis of milk components and linked to MG metabolism did not differ between CONT and REST (Figure 1).

### 2.3. Microarray Analysis

Mammary gene expression analyzed by a microarray assay allowed the identification of 33 differentially expressed genes, including 25 known genes (corrected *p_adj_* < 0.05), between CONT and REST, 24 h after the inflammatory challenge by LPS (Table 1). The expression increased for 19 and decreased for 6 genes in REST compared with CONT. All these DEG presented a fold change (FC) greater than 1.4, with two genes (*PDK4* and *SLC25A34*) presenting an FC greater than 4. Gene ontology and function analyses revealed that most DEG are involved in metabolism, including the regulation of fatty acid (FA) oxidation, glucose, and protein metabolism, and in immune responses (Figure 2). The results obtained, using Pathway Studio^®^ software, were consistent with those from Panther software. We focused on the most represented functions, in particular, those involved in metabolism and immune response. We identified DEG in FA and glucose metabolism (*CPT1A*, *PDK4*, *PFKFB4*), carnitine shuttle (*SLC25A20*, *CPT1A*, *SLC25A34*), regulation of cellular ketone metabolic process (*PDK4*), and the key genes in those processes. A number of genes involved autophagy (*PFKFB4*, *DNNED*) and immune function (*PGLYRP3*, *KLF13*, *PLEKHA2*, *WC7*, *TRIB2*, *CXCR7*, and *MBP*) processes also were altered.

### 2.4. LC MS/MS Proteomic Analysis

We identified 1475 proteins, 967 of which were validated with more than two peptides and considered for further investigation. Fifty three proteins were differentially expressed proteins, 10 were upregulated and 43 were downregulated in REST (*p* < 0.05; Table 2; Appendix A). Classification of DEP highlighted proteins involved in the immune process, metabolism (regulation of protein catabolic and carbohydrate metabolic processes) and cell functioning (such as RNA splicing, translation, or regulation of cell adhesion). Proteins involved in apoptosis are also identified (Figure 3). DEP are involved in protein folding and post-translational modifications (GANAB, PDIA3, RPN2, RPN1, CCT4, PDIA4, and PPIB), protein catabolic process (PSMD2 and PPP2CA), carbohydrate metabolism (PAPSS1, RPS27A, GANAB), synthesis of immunoglobulins (F1MH40, F1MLW8), and regulation of inflammatory response (PDIA3, PSMA3, PSMD2, PPP2CA, RPS2, RPL10, RPS15, CASP6, PCBP2, and STAT5A). Many DEP are involved in RNA splicing (HNRNPH1, DHX9, HNRNPC, YBX1, PCBP2, SNRPA1, HNRNPA3, and PPP2CA) or translation processes (RPS27A, RPL10, RPS15, RPS2, EIF3H, RPN2, RPN1, FARSB, and NACA). Additional analysis by Uniprot software revealed similar classifications confirming them.

## 3. Discussion

This study assesses the effects of undernutrition and the resulting metabolic imbalance on the mammary gland (MG) inflammatory response in early lactation cows using a nutrigenomic approach. The effects of dietary treatments are confirmed by decreased intake, milk yield, energy balance in underfed (REST) cows, and by changes in blood metabolite and insulin concentrations (an increase in plasma NEFA and BHBA and a decrease in insulin and glucose concentrations; Supplementary Appendix A; [25]). The inflammatory response to intra-mammary lipopolysaccharide is confirmed by clinical parameters, such as milk SCC, rectal temperature, and other classical clinical symptoms [25]. The effects of nutrient restriction and metabolic imbalance on the inflammatory response, at the RNA and protein levels, were evaluated using transcriptomic and proteomic analyses. These analyses were performed using MG samples were obtained by biopsies performed 24 h after the LPS challenge. We performed a single biopsy to avoid the potential interference of repetitive biopsies on the inflammatory response. Also, the adjacent quarter may not constitute a good control for the LPS challenged quarter, as inflammation cytokines may exert local effects and influence adjacent quarters [19]. However, one limitation of this design is that the present experimental design does not allow a kinetic data to follow the establishment of inflammation.

### 3.1. Gene Expression Changes at mRNA Level

The study at the mRNA level was performed using two complementary approaches. The first one was a targeted approach to focus our attention on the inflammatory response, then a global analysis was performed using microarray study. We investigated the effects on candidate genes by RT-qPCR, the majority, which are involved in the immune response of interest for inflammation. The expression of candidate genes did not differ between REST and with the control (CONT) group (Figure 1). This result suggests that the expression of genes considered important in the inflammatory response [11,26,27,28] are not altered by the nutrient restriction in MG 24 h after LPS administration. This experimental design does not allow the ability to evaluate the modification of the expression of these genes during early inflammatory response. A tendency for decreased expression of the *TAP* gene in REST was observed. The product of *TAP* gene is a member of the family of small cationic peptides that have widespread antimicrobial activity; TAP is expressed by bovine mammary epithelial cells [29] and has a broad-spectrum activity against different strains of bacteria, including *E. coli* [30]. The upregulation of *TAP* gene expression in REST may constitute a protective mechanism against pathogens.

To complete the candidate gene analyses, a global gene expression approach, using a bovine microarray was used to assess the molecular mechanisms underlying metabolic and inflammatory MG responses, potentially affected by undernutrition and negative energy balance. Transcriptomic analysis revealed 33 differentially expressed genes in MG 24 h after LPS challenge in REST compared to CONT. The number of DEG detected in our study are small compared to the research assessing the effects of inflammation on MG transcriptome [18,19,31]. This study does not compare normal versus inflamed MG, rather it aims to evaluate the effects of undernutrition during the inflammatory response, and therefore both REST and CONT were challenged with LPS. The DEG were classified in six functional classes. In this discussion, we mainly focus our attention on genes that play a role in the metabolic processes (FA, glucose and protein metabolism) and on those involved in immune function.

#### 3.1.1. DEG Involved in Metabolism

The classification of genes by bioinformatics analyses indicate that metabolic process (FA oxidation, glucose, and protein metabolisms) is the class most altered by undernutrition after an LPS challenge conditions. Among the genes presenting the highest fold change (between 2.7 and 6.8) are *PDK4*, *CPT1*, and *SLC25A34*, which are involved in glucose and FA metabolism. *PDK4* plays a key role by inhibiting the pyruvate dehydrogenase complex. This inhibition prevents the formation of acetyl-coenzyme A from pyruvate [32], resulting in an expected decrease in glucose and an increase in fat utilization in response to prolonged undernutrition [33]. The upregulation of *PDK4* was also found in leucocytes of underfed ewes, however, its expression was downregulated during an intra-mammary inflammatory challenge [34]. The large increase of *PDK4* expression in REST is in agreement with decreased insulinemia and with the upregulation of *ESRRA*. The gene *ESRRA* upregulates *PDK4* expression [35]. Both genes spare glucose and promote FA β-oxidation, therefore, their upregulation in REST would allow MG to shift the metabolic pathways from glycolysis to β-oxidation. The upregulation of *CPT1* gene expression would also promote the β-oxidation [36,37], since it is a rate-limiting step of FA entry in mitochondria [38]. This is in line with the increased expression of the *CPT1* gene observed in whole blood transcriptome of underfed dairy sheep [34]. The increase in β-oxidation is further supported by an upregulation of *SLC25A20* and *SLC25A34* in REST, which are two members of *SLC25* mitochondrial carrier family. *SLC25A20* transports carnitine and carnitine-FA complexes across the inner mitochondrial membrane. *SLC25A34* is supposed to act in a similar way, but its exact function still is not known totally [39]. MG seems to spare glucose (downregulating glycolysis) and promote FAs as an energy source (upregulating β-oxidation; Figure 4) in order to adapt to underfeeding. The mammary expression of genes involved in lipid metabolism is also modified in comparison with NEB (induced by caloric restriction) and the positive energy balance of cows after the peak in lactation [40]. Interestingly, the downregulation of genes linked to fat metabolism (FA biosynthesis) is observed 24 h after *E. coli* infection in MG of lactating cows [18]. This suggests that inflammation downregulates the FA biosynthesis and may increase the use of preformed FA, derived from other sources, such as from the mobilization of adipose tissue.

These results suggest that energy metabolism modifications, in response to inflammation, are more marked in REST than in CONT, probably due to the expected limited availability of nutrients to support an acute inflammation in REST.

#### 3.1.2. DEG Involved in Immune Response

The objective of this study was to identify the effects of an aggravated NEB on MG transcriptomic responses to inflammation. The experiment was not designed to compare gene expression in normal versus LPS challenged MG, which have been previously reported [2,11,18,31]. Microarray analysis shows that, the immune response together with the metabolic processes, are the main biological processes modified by undernutrition after LPS challenge (Figure 2), with modifications of genes different from those selected for RT-qPCR analyses. However, immune response was not the main biological process to be affected by restriction during inflammation. Among the upregulated genes is *PGLYRP3*, which belongs to a family of innate immunity pattern recognition molecules that are activated by LPS and bactericidal and bacteriostatic properties and are activated by LPS [41]. When the invading bacteria survive, neutrophil infiltration is replaced with T and B lymphocytes and monocytes [42]. The upregulation in REST of *KLF13*, *PLEKHA2*, *WC7*, and *MBP*, involved in the immune response by activating T and B lymphocytes, therefore, is in line with the expect recruitment of leukocytes by MG (Figure 5). Additionally, the upregulation of *PLEKHA2* in REST, a gene involved in the cell adhesion process [43], suggests that there is an increased migration of B leucocytes. Together, the upregulation of these genes suggests a different nature or a higher response to LPS stimuli in REST compared with CONT. Nevertheless, the deregulation of *TRIB2* and *CXCR7* suggests that the immune process could be impaired. Indeed, both genes participate in the activation of immune cells and influence IL-8 production, a chemokine upregulated in response to infection [10], where the concentration increased in milk, within 4 h after LPS infusion [(25], and within 16–24 h after experimental infection with different strains of *E.coli* or LPS infusion [44,45]. The upregulation of *TRIB2* and the downregulation of *CXCR7* genes, however, suggests a potential IL-8 production alteration in response to inflammation in REST. These results contrast with the expected inflammatory response and could be a sign of deficient immune function under exacerbated NEB. Taken together, differences in gene expression might suggest a modified resolution of inflammation in response to LPS, due to aggravated NEB. During the course of an experiment with LPS, the inflammatory response usually declines within 24 h [44]. The REST cows might have experienced difficulties in restoring the MG homeostasis by 24 h after LPS challenge, due to the metabolic changes inherent in nutrient deficiency. However, this conception needs more detailed investigation, with a kinetic analysis, to be confirmed.

### 3.2. Gene Expression at Protein Level

#### 3.2.1. Proteins Involved in Protein Synthesis

Among the 53 differentially expressed proteins (DEP) in REST compared to CONT, 43 were downregulated. Most of these are involved in RNA and protein metabolism, with roles that vary from RNA splicing to translation. The downregulation of proteins involved in the splicing process, such as HNRNPH1, HNRPC, HNRNPA3, PCBP2, YBX1, SNRPA1, and DHX9, suggests that splicing is impaired in the MG of REST cows. This could explain in part the reduced synthesis and secretion of milk protein in REST compared with CONT [25]. Moreover, altered splicing and translation mechanisms might have a profound influence on protein biochemical properties and, ultimately, alter immune response to pathogens. Among the four proteins belonging to the HRNPs family (HNRNPH1, HNRPC, HNRNPA3, PCBP2), the first three are RNA binding proteins associated with pre-mRNAs in the nucleus, influencing pre-mRNA processing as well as other aspects of mRNA metabolism and transport. The dysfunction of HRNPs is linked to different proliferative and degenerative diseases [46], but the role of these proteins in the inflammatory response is still not fully understood. Some members of this family are reported to ensure resolution of inflammation [47]. However, the role of HRNPs could be associated with a mammary remodeling due to the restriction. In our study, YBX1 and DHX9 are downregulated in REST and the downregulation of these two genes is associated with impaired inflammatory responses [48]. Additionally, the loss of PP2AC function causes severe immunological disorders in Treg cells [49]. Thus, the downregulation of all these proteins in REST (Figure 6) suggest a modified inflammatory response in underfed early lactation cows.

A number of proteins involved in translation are downregulated in REST (Figure 6; RPS27A, RPS15, RPS2, RPL10, EIF3H, RPN2, RPN1, FARSB, and NACA). Four riboprotein family members (3 RPS and 1 RPL) are part of a ribosome. Interestingly, protein building ribosomes alone are shown to affect the other cell processes outside the ribosome like development, apoptosis, and aging during their altered expression levels [50]. Additionally, the decrease of RPN1 and RPN2, which catalyze co-translational N-glycosylation, suggests an impaired post-translational protein modification process in the REST group. This process may play an important role in the immune system by creating the glycans on an immune cell′s surface that helps migration of the cell or by glycosylating the various immunoglobulins [51]. Elsewhere, it is reported that this process can be defective during glucose deficit, leading to a reduction of protein glycosylation and harmful accumulation of unfolded proteins [52]. The decrease in RPN1 and RPN2, observed in the current study, could potentially lead to the creation of misfolded proteins in MG of REST cows.

Additionally, protein folding and its control might be modified in REST, due to the downregulation of chaperone proteins such as PDIA3, PDIA4, and CCT4 (Figure 6). PDIA3 and PDIA4 are part of a larger super-family of a disulfide isomerase family of endoplasmic reticulum proteins that catalyze protein folding [53]. PDIA3 contributes to the correct folding of glycoproteins [54]. The loss of PDI activity and the consequent accumulation of misfolded proteins are associated with chronic inflammation [54,55]. Moreover, PDIA3 is a structural component required for the stable assembly of the peptide-loading complex of the major histocompatibility complex class I pathway. Its activity seems to play a role in lymphocyte T and B function [56]. Added to its role in folding, PDIA4 promotes the immunoglobulin G intermolecular disulfide bonding and antibody assembly in vitro [57]. Because CCT4 assists in the folding of newly translated polypeptides, this function might have been altered in REST [58]. Overall, proteomic data strongly suggest that protein synthesis is impaired by undernutrition at different levels (translation, folding, and post-translation modifications). The modifications of protein metabolism might partially explain the lower milk protein yield from 1.12 to 0.62 kg/ observed during restriction.

#### 3.2.2. DEP Involved in Immune Response

Undernutrition downregulated FARSB protein expression. The decrease of this protein is linked with impaired acute inflammation responses in mice [59], suggesting an impairment in immune system function. In contrast, there was an upregulation of proteins such as SERPINA3, SERPINA3-5, F1MLW8, and Q1RMN8. SERPINA is an acute-phase protein, whose concentration can rise during acute and chronic inflammation [60]. F1MLW8 and Q1RMN8 proteins are similar to immunoglobulin lambda and typical for B-cells, and are important for its maturation from pre-B cells to mature ones [61]. The increase of these four proteins in REST MG, suggests that the resolution inflammation process was delayed in REST, compared to CONT 24 h after LPS challenge, whereas it could be considered that, potentially, it has already resolved in CONT. This is in line with the reported peak in SCC at 12 h that declines 24 h after LPS challenge [44].

The decreased translation process and post-translational protein modification (folding and glycosylation), that is observed at the protein level, might alter protein synthesis and activate an unfolded protein response [52]. This role could also be suggested to affect the proteins involved in the immune response.

## 4. Materials and Methods

### 4.1. Ethics Statement, Treatments and Sampling

The cows were housed at the Herbivore Research Unit of INRA Research Center of Auvergne–Rhone–Alpes. Animal procedures were performed in compliance with Regional Animal Care Committee guidelines CEMEAA: Auvergne, French Ministry of Agriculture and European Union guidelines for animal research C2EA-02. All procedures were approved by the regional ethics committee on animal experimentation (APAFIS #2018062913565518). The animals were in their second to the fourth days of lactation, with a body condition score (BCS) of 2.0 to 2.2 (0 to 5 scale), a week before feed restricted diet. All animals were observed for uterine disease and did not present any signs of abnormality. Additionally, the health history of each animal was inspected and only those without any health problems, within the last 6 months before calving, were chosen.

At 24 ± 3 days in milk, sixteen multiparous Holstein cows were allowed *ad libitum* intake of a lactation diet CTRL, *n* = 8, 7.1 MJ/kg DM NEL, 17.4% Crude Protein. Their diet constituted of corn (24.2% dry matter), corn silage (29%), grass silage (25.5%), soybean meal (16.9%), and complemented with vitamins and minerals (0.9%). The underfed (REST) group received a ration diluted with barley straw (48% DM) for 96 h (RES, *n* = 8; 5.16 MJ/kg DM NEL, 12.2% CP). Therefore, the ratio of forage to concentrate differed from 58.0/42.0 in control (CONT) group to 79.2/20.8 in REST group [25]. Dry matter intake, milk yield, energy balance, plasma insulin, glucose, non-esterified fatty acids (NEFA) and BHB (β-hydroxybutyrate) concentrations did not differ between CONT and REST immediately before underfeeding (21.8 kg/day, 39.0 kg/day, –5.6 MJ/day, 22 µIU/mL, 3.78, 0.415, and 0.66 mM, respectively, at day –1), but were significantly altered in REST at 72 h of underfeeding (Appendix A). Following 72 h of restriction or control diet, one healthy rear mammary quarter was injected with 50 µg of lipopolysaccharide *E. coli* 0111:B4; (LPS-EB Ultrapure, InvivoGen, San Diego, CA, USA) diluted in 10 mL of sterile saline (CDM Lavoisier, Paris, France), containing 0.5 mg/mL BSA cell culture grade, endotoxin free, A9576, (Sigma–Aldrich, St. Louis, MO, USA), using a sterile disposable syringe fitted with a sterile teat cannula. Mammary biopsies were performed 24 h after the LPS injection, as previously described [62], corresponding to 96 h of feed restriction or control diet. Tissue samples were immediately frozen in liquid nitrogen and stored at −80 °C prior to RNA and protein analyses.

Throughout the study, milk samples were collected at 4 consecutive milkings each week before the beginning of the restriction and just before the LPS challenge and analyzed for SCC. Only healthy cows were included in the study. Additionally cows were screened for mastitis, one week before and immediately before the LPS challenge, using the California Mastitis Test (Neodis, Rambouillet, France) for all quarters, and somatic cell counts of rear quarter milk samples (Galilait, Theix, 63122 Saint Genès–Champanelle, France), one week before and immediately prior to the LPS challenge. Only cows with SCC lower than 100,000 cells/mL, in a rear quarter, were included in the study. Indeed, cows were considered healthy if the quarter SCC was inferior to 100,000 cells/mL and were free of any other signs of health problems [25]. Additionally, foremilk samples were collected from the LPS challenged quarters, immediately before the morning milking that preceded the LPS injection (time 0), and at 4, 6, 10, and 24 h after LPS injection. These quarter milk samples were analyzed for IL-8, IL1-β, TNF-alpha, and CXCL3 using Elisa [25].

### 4.2. RNA Preparation and Analyses

RNA and protein extractions were performed from the same mammary biopsy samples *n* = 16 animals, (8 CONT and 8 REST). The total RNA was extracted from 50 mg of the mammary gland (MG) by using the mirVana miRNA Isolation Kit (Thermo Fisher Sciences, Waltham, MA USA). The concentration and purity of RNA were estimated by spectrophotometry NanodropTH, (ND-1000, NanoDrop Technologies LLC, Wilmington, DE, USA), and by using the Bioanalyzer 2100 (Agilent Technologies Inc., Santa Clara, CA, USA), respectively. Once these validation steps were completed, only 12 cows (6 RES and 6 CTR) were kept for gene expression analyses at mRNA level, which presented a good and uniform quality of the samples for a microarray experiment.

### 4.3. RT-PCR Analyses

Reverse transcription (RT) was performed on 2 μg of total RNA using the “High Capacity RNA to cDNA” kit and following the manufacturer’s recommendations (Applied Biosystems, Villebon Sur Yvette, France) in a final volume of 20 μL. In parallel, negative controls were performed without the matrices. The primers are described in Table 3. The genes, *UXT2*, *CLN3*, and *EIF3K* were used as housekeeping genes [63]. Real-time quantitative PCR was performed on the StepOnePlus™ PCR System (Applied Biosystems, Villebon Sur Yvette, France), using 5 μL of 50 fold-diluted single-stranded cDNA and the TFPower SYBRGreen PCR Master Mix, according to the manufacturer’s instructions (Applied Biosystems, Villebon Sur Yvette, France). Subsequent to an initial denaturing step (95 °C for 10 min), the PCR mixture was subjected to the following two-step cycle, which was repeated 40 times: Denaturing for 15 s at 95 °C and annealing and extension for 45 s at 60 or 62 °C. The results were expressed as a fold change of Ct values relative to the control using the Δ^Ct^ method [64]. The significance was determined using a *t*-test with *p* < 0.05 considered as significant.

### 4.4. Microarray Analyses

Microarray analyses were performed on twelve animals (6 RES and 6 CTRL) using 100 ng of total RNA from each MG sample. The total RNA was amplified, fluorescently labeled, and hybridized to the bovine 4 × 44K microarray (Agilent Technologies, Inc. Santa Clara, CA, USA), and all the procedures described below were performed according to the manufacturer instructions (Agilent Technologies, Inc. Santa Clara, CA, USA). Briefly, for each hybridization, the total RNA was linearly amplified and labeled with Cy3 using the one-color Low Input Quick Amp Labeling Kit. Then, 1650 ng of Cy3-labeled cRNA was hybridized on the microarrays using the Gene Expression Hyb Kit. Hybridization was performed for 17 h at 65 °C in a rotating hybridization oven at 10 rpm. Following hybridization, all microarrays were washed and scanned using the Agilent Microarray Scanner G2565A. The resulting TIFF-images (Tagged Image File Format) were processed using Feature Extraction software Version 11 to obtain normalized data. Normalized with 75th percentile shift, the data were analyzed using GeneSpring software. The moderated *t*-test with Westfall–Young familywise error rate (FWER) correction was applied [70]. The differences were considered significant at an adjusted *p* < 0.05. The data were accessible through the GEO series accession number GSE114975. The classification and functional analyses of differentially expressed genes were performed using PANTHER [71] and confirmed using Pathway Studio^®^ software (Elsevier, The Netherlands).

### 4.5. Protein Preparation and Analyses

The proteins were extracted by homogenizing 80 mg MG tissue (*n* = 16; 8 RES and 8 CTRL) in 2 mL lysis buffer (8.3 M urea, 2 M thiourea, 2% CHAPS, 1% DDT). Following homogenization, the samples were incubated for 5 min at room temperature and centrifuged at 10,000× *g* for 30 min at 8 °C. The protein concentrations were measured in supernatant with Quick Start Bradford protein assay (BioRad, Marnes–La–Coquette, France), aliquoted and then stored at −20 °C, until further preparation. Sample supernatants were mixed with 1 volume of Laemmli buffer and heated at 60 °C for 5 min. Separation, by SDS-PAGE (12% acrylamide), was performed using a Mini-Protean II electrophoresis unit (BioRad, Marnes–La–Coquette, France) and 100 μg protein loaded per lane. To concentrate the samples, the gels were run at 80 V until the dye front reached the bottom of the concentration gel. Gels were stained overnight in Coomassie brilliant blue G-250. Excised lanes were reduced and alkyled before de-staining in 25 mM ammonium bicarbonate with acetonitrile (50/50 *v*/*v*). Following dehydration with 100% acetonitrile, gel pieces were dried in a Speed Vacuum and the samples were preserved at −20 °C until LC MS/MS analysis.

### 4.6. LC MS/MS Analysis

The proteins were hydrolyzed overnight at 37 °C, using 800 ng (80 µL) of sequence grade-modified trypsin (Promega, France) per band. Subsequent to extraction by 64 µL of acetonitrile 100% and sonication, the peptides were concentrated in a Speed Vacuum and volume was adjusted to 30 µL with an aqueous solution (99.9% H2O, 0.1% TFA). Peptide mixture (2.5 µL) was injected into the nano HPLC Ultimate 3000, (Thermo Fisher Scientific, Courtaboeuf, France) after a preliminary step of desalting and concentration in the pre-column 300 µm × 5 mm, (ThermoFisher, Courtaboeuf, France) for 6 min, and a second step of separation in an analytical C18 column 75 µm, 25 cm, (Pepmap Thermo Fisher Scientific, Courtaboeuf, France) with a 10–40% gradient (A: 0.1% FA in water, B: 0.1% FA in acetonitrile) at 450 nL/min. The eluate was electrosprayed through the CaptiveSpray ion source into the mass spectrometer QTOF impact II (Bruker, Wissembourg, France) operated in CID Data Dependent mode. Each MS analysis was succeeded by as many MSMS analysis as possible within 3 s.

### 4.7. Protein Identification and Label-Free Quantitation

The raw files were loaded, at the end of each LC-MS/MS analysis, into the Progenesis QI software Non-linear Dynamics, v 4.1 (Newcastle upon Tyne, UK) and label-free quantitation was performed using a proprietary workflow alignment, peak picking, normalization, design set up, quantitation, and protein identification.

Regarding protein identification (Appendix A), the Mascot V.2.5, internally licensed version (www.matrixscience.com) was used with uniprot-ref_*Bos taurus* database 19.840 sequences (07/2015). The following parameters were considered for the searches: Peptide mass tolerance was set to 10 ppm; fragment mass tolerance was set to 0.05 Da and a maximum of two missed cleavages was allowed. Variable modifications were methionine oxidation (M), carbamidomethylation (C) of cysteine and Deamidated (NQ). Protein identification was considered valid, if at least two peptides with a statistically significant Mascot score were assigned, with a false discovery rate (FDR) less than 1%.

Concerning label-free quantitation, all unique validated peptides of an identified protein were included, and the total cumulative abundance was calculated by summing up the abundances of all unique peptides allocated to the respective protein. Statistical analysis was performed, using the “between subject design,” and the *p*-values were calculated by an analysis of variance, using the normalized abundances across all runs. Differential proteins were conserved for interpretation if the peptides’ individual abundances showed a good correlation with protein abundance. All differential proteins were inspected manually with these correlation criteria. To extract the maximum biological information of differentially expressed proteins, PANTHER [71], Pathway Studio^®^ software (Elsevier, The Netherlands) and UniProt [72], were used.

## 5. Conclusion

Undernutrition affected multiple aspects of MG function, as demonstrated by modifications of milk secretion, and MG mRNA and protein expression. During this study, expression analyses were performed 24 h post-LPS challenge corresponding to the period of inflammation resolution. The effects of undernutrition on studied candidate genes, known as major genes relating to the innate immune responses, were weak. Therefore, the transcriptomic and proteomic analyses pointed out modifications of energy metabolism (fatty acid and glucose), and protein metabolism (synthesis and post-translational modification), respectively, but relatively few genes involved in immune response were affected. Our nutrigenomic analyses have suggested that undernutrition of early lactating cows modified the mammary gland metabolism. The holistic analyses of the systemic reaction in the mammary gland expands the knowledge of the effects of NEB and metabolic imbalance occurring in early lactation, during inflammation. These identified genes may be relevant for quantitative trait loci studies and genomic selection.

## Figures and Tables

**Figure 1 ijms-20-01156-f001:**
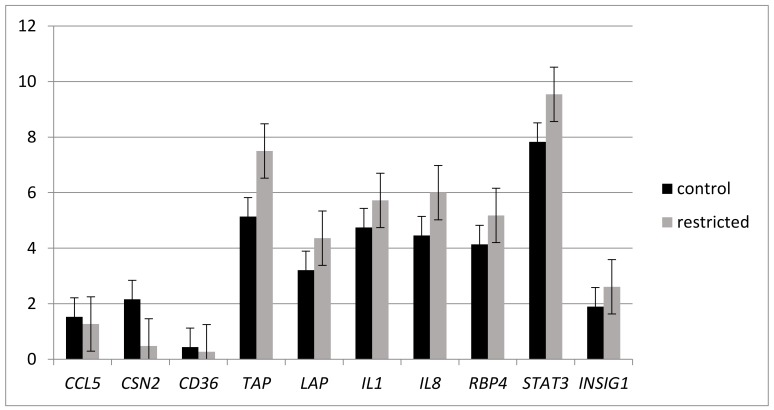
Effects of nutrient restriction and intra-mammary lipopolysaccharide (LPS) challenge on gene mRNA expression quantified by RT-qPCR and presented as ∆^CT^. Comparison of the gene expression does not show a difference between control (CONT; *n* = 6) and restricted (REST; *n* = 6) Holstein cows (*p* ≥ 0.1). The expression of the *TAP* gene tended to differ (*p* = 0.07). *UXT2*, *CLN3,* and *EIF3K* were used as housekeeping genes.

**Figure 2 ijms-20-01156-f002:**
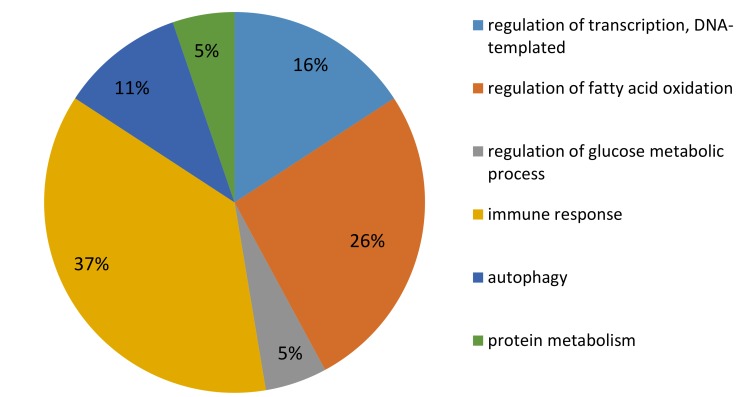
Main biological processes of differentially expressed genes in mammary glands (MG) of underfed (REST) versus control (CONT) early lactation cows during an acute inflammation identified by transcriptomic analysis. Each percentage was indicated for each biological processes. Bioinformatics analyses were performed using Panther and Pathway Studio^®^ software.

**Figure 3 ijms-20-01156-f003:**
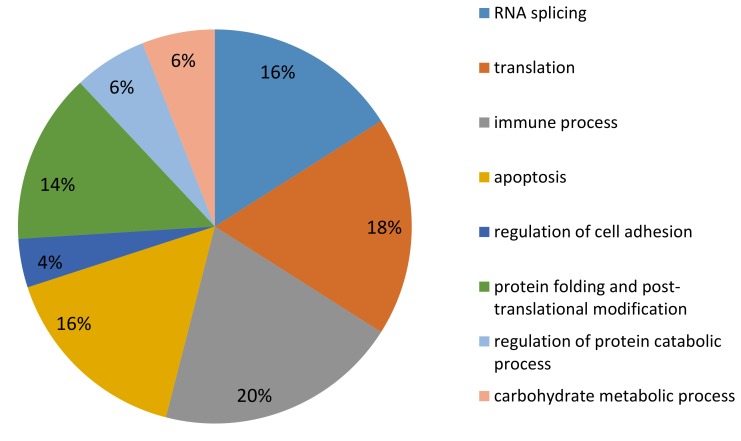
Main biological processes of differentially expressed proteins in MG of underfed (REST) versus control (CONT) early lactation cows during acute inflammation. Each percentage was indicated for each biological processes. Bioinformatics analyses were performed using Uniprot and Pathway Studio^®^ software.

**Figure 4 ijms-20-01156-f004:**
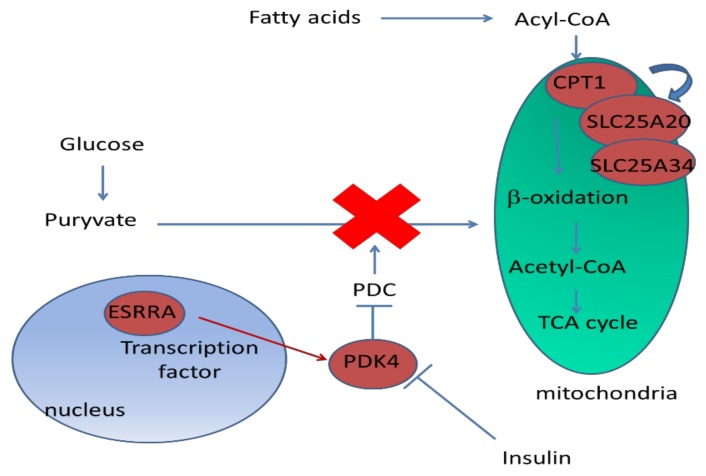
Upregulation of genes detected at mRNA level highlights a potential increase of fatty acid oxidation. Red boxes indicate upregulated genes. Blue and green boxes represent the nucleus, and mitochondria, respectively. *Carnitine O-palmitoyltrasferase 1 (CPT1A)* encodes carnitine O-palmitin O-palmitoyltranferase 1, *Mitochondrial Carnitine/Acylcarnitine Carrier Protein (SLC25A20)* and *Solute Carrier Family 25 Member 34 (SLC25A34)* are two members of the solute carrier family 25. *Steroid Hormone Receptor ERR1 (ESRRA) and Pyruvate Dehydrogenase Acetyl-Transferring Kinase Isozyme 4 (PDK4)* encode a steroid hormone receptor and a pyruvate dehydrogenase, respectively.

**Figure 5 ijms-20-01156-f005:**
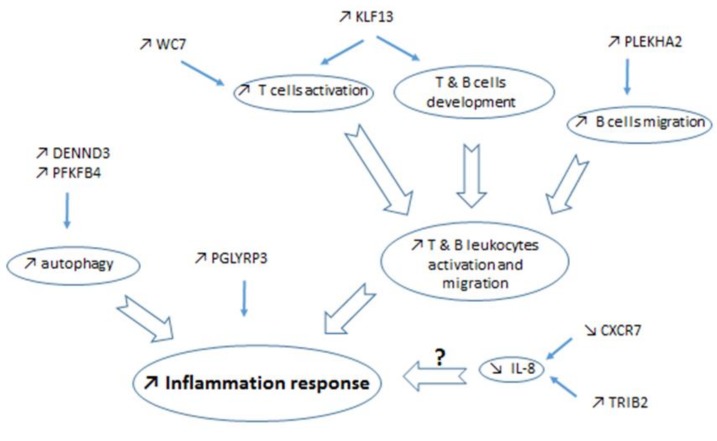
The potential actions of the differentially expressed genes identified in the comparison of mammary transcriptomes of restricted versus control cows during LPS challenge were from bioinformatics analyses and literature.

**Figure 6 ijms-20-01156-f006:**
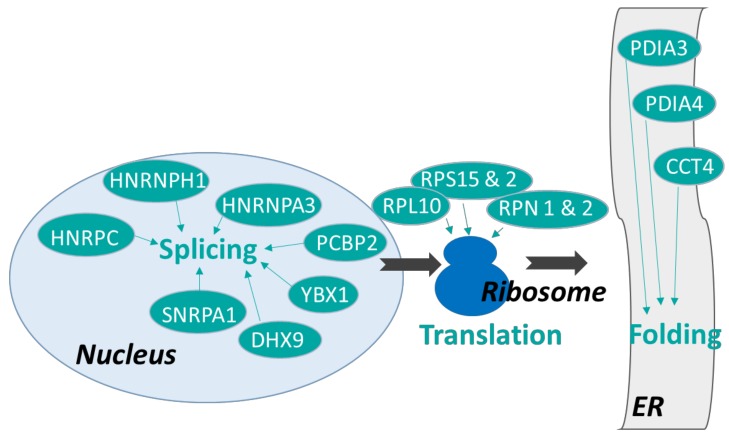
Identified differentially expressed genes are presented in their potential role in crucial steps of the control of gene expression. Green boxes indicate genes which are downregulated by nutritional restriction. Light and dark blue represent the nucleus and ribosomes, respectively. Grey boxes represent the endoplasmic reticulum. Differentially expressed proteins between restricted and control cows after four days of undernutrition for restricted cows during LPS challenge.

**Table 1 ijms-20-01156-t001:** Differentially expressed genes (alphabetic classification) in MG of early lactation Holstein cows in response to undernutrition and intra-mammary lipopolysaccharide (LPS) challenge. Transcriptomic comparison of control (CONT; *n* = 6) and restricted diet (REST; *n* = 6). Normalized microarray data were analyzed using GeneSpring software and moderated *t*-tests with Westfall–Young correction. *p*_adj_ ≤ 0.05 and ranged from 0.01 to 0.04. FC were >1.4.

Gene Symbol	Accession Number	Gene Name	Fold Change
*PDK4*	NM_001101883.1	*[Pyruvate Dehydrogenase (Acetyl-Transferring)] Kinase Isozyme 4, Mitochondrial*	6.80
*SLC25A34*	NM_001034497.2	*Solute Carrier Family 25 Member 34*	4.63
*PGLYRP3*	XM_010826801.2	*Peptidoglycan Recognition Protein 3*	3.31
*CPT1A*	NM_001304989.1	*Carnitine O-Palmitoyltransferase 1*	2.77
*NEFL*	NM_174121.1	*Neurofilament Light Polypeptide*	2.76
*KNTC1*	NM_001192091.	*Kinetochore-associated protein 1*	2.58
*MBP*	NM_001206674.1	*Myelin basic protein*	2.37
*DENND3*	XM_010822660.2	*DENN domain-containing protein 3*	2.33
*TFAP2D*	NM_001192329.1	*Transcription factor AP-2-delta*	2.29
*PFKFB4*	NM_001192835.1	*6-Phosphofructo-2-kinase/Fructose-2,6-bisphosphatase 4*	2.04
*TRIB2*	NM_178317.3	*Tribbles homolog 2*	2.03
*WC-7*	NM_001281912.1	*WC1 isolate CH149*	1.85
*TMEM50B*	NM_001034786.2	*Transmembrane protein 50B*	1.78
*ALAD*	NM_001014895.1	*Delta-aminolevulinic acid dehydratase*	1.77
*LOC517144*	XM_002699133.1	*Putative olfactory receptor 10D3*	1.74
*ESRRA*	NM_001191373.2	*Steroid hormone receptor ERR1*	1.73
*SLC25A20*	NM_001077936.2	*Mitochondrial carnitine/Acylcarnitine carrier protein*	1.70
*PLEKHA2*	NM_001035383.1	*Pleckstrin homology domain-containing family A member 2*	1.69
*KLF13*	NM_001083533.1	*Krueppel-like factor 13*	1.67
*ARMC1*	NM_001015594.2	*Armadillo repeat-containing protein 1*	−1.43
*RPL-37A*	NM_001035008.2	*Similar to 60S ribosomal protein L37a*	−1.76
*BBS9*	NM_001192853.1	*Protein PTHB1 / Bardet-bield syndrom 9*	−1.79
*SLC9A7*	XM_015470441.1	*Sodium/Hydrogen exchanger 7*	−1.99
*CXCR7*	NM_001098381.2	*Atypical chemokine receptor 3*	−2.10
*OOEP*	NM_001077869.2	*Oocyte-expressed protein homolog*	−2.52

**Table 2 ijms-20-01156-t002:** The list of differentially expressed proteins (up- and down-regulated then alphabetic classification) 24 h after inflammation challenge by LPS in MG of underfed (REST; *n* = 6) compared with control (CONT; *n* = 6) cows. Proteins were analyzed with Progenesis LC-MS software v4.1 (Nonlinear Dynamics). The minimum mascot score validation for one peptide was 31 with a rate of false discovery <1%.

Gene Symbol	Accession Number	Protein Name	Fold Change
Unknown	F1MLW8_BOVIN	Uncharacterized protein / Immunoglobulin light chain, lambda	5.5
*SERPINA3*	G8JKW7_BOVIN	Uncharacterized protein	2.4
*SERPINA3-5*	SPA35_BOVIN	Serpin A3-5	2.0
Unknown	Q1RMN8	TREMBL:Q1RMN8 (Bos taurus) Similar to Immunoglobulin lambda-like polypeptide 1	1.9
*ALDH18A1*	Q2KJH7_BOVIN	Aldehyde dehydrogenase 18 family, member A1	1.7
Unknown	F1MH40_BOVIN	Uncharacterized protein	1.5
*HBA*	HBA_BOVIN	Hemoglobin subunit alpha	1.3
*HBB*	HBB_BOVIN	Hemoglobin subunit beta	1.3
*PCBP2*	Q3SYT9_BOVIN	Poly(RC) binding protein 2	1.2
*ARPC2*	ARPC2_BOVIN	Actin-related protein 2/3 complex subunit 2	1.1
*CCT4*	TCPD_BOVIN	T-complex protein 1 subunit delta	−1.2
*DDX17*	A7E307_BOVIN	DDX17 protein	−1.2
*GANAB*	F1N6Y1_BOVIN	Uncharacterized protein	−1.2
*HNRNPH1*	E1BF20_BOVIN	Uncharacterized protein	−1.2
*HNRPC*	Q3SX47_BOVIN	Heterogeneous nuclear ribonucleoprotein C (C1/C2)	−1.2
*LMAN2*	A6QP36_BOVIN	LMAN2 protein	−1.2
*NACA*	NACA_BOVIN	Nascent polypeptide-associated complex subunit alpha	−1.2
*PDIA4*	PDIA4_BOVIN	Protein disulfide-isomerase A4	−1.2
*PPP2CA*	PP2AA_BOVIN	Serine/threonine-protein phosphatase 2A catalytic subunit alpha isoform	−1.2
*PSMA3*	PSA3_BOVIN	Proteasome subunit alpha type-3	−1.2
*RPS2*	RS2_BOVIN	40S ribosomal protein S2	−1.2
*RPS27A*	RS27A_BOVIN	Ubiquitin-40S ribosomal protein S27a	−1.2
*HNRNPA3*	E1BEG2_BOVIN	Uncharacterized protein	−1.1
*PDIA3*	PDIA3_BOVIN	Protein disulfide-isomerase A3	−1.1
*PRKCDBP*	PRDBP_BOVIN	Protein kinase C delta-binding protein	−1.1
*ACTR1A*	F2Z4F0_BOVIN	Uncharacterized protein	−1.3
*DHX9*	DHX9_BOVIN	ATP-dependent RNA helicase A	−1.3
*GNB1*	GBB1_BOVIN	Guanine nucleotide-binding protein G(I)/G(S)/G(T) subunit beta-1	−1.3
*KIF5B*	F1N1G7_BOVIN	Kinesin-like protein	−1.3
*PLBD2*	PLBL2_BOVIN	Putative phospholipase B-like 2	−1.3
*RPN1*	A3KN04_BOVIN	Dolichyl-diphosphooligosaccharide--protein glycosyltransferase subunit 1	−1.3
*SNRPA1*	A6H788_BOVIN	SNRPA1 protein	−1.3
Unknown	F6PWD5_BOVIN	Uncharacterized protein (Fragment)	−1.3
*CAPZA2*	CAZA2_BOVIN	F-actin-capping protein subunit alpha-2	−1.4
*FARSB*	A8E4P2_BOVIN	FARSB protein	−1.4
*HSPA4*	E1BBY7_BOVIN	Uncharacterized protein	-1.4
*PAPSS1*	Q3T0J0_BOVIN	3′-phosphoadenosine 5′-phosphosulfate synthase 1	−1.4
*PPIB*	PPIB_BOVIN	Peptidyl-prolyl cis-trans isomerase B	−1.4
*PSMD2*	PSMD2_BOVIN	26S proteasome non-ATPase regulatory subunit 2	−1.4
*RPS15*	RS15_BOVIN	40S ribosomal protein S15	−1.4
*STAT5A*	STA5A_BOVIN	Signal transducer and activator of transcription 5A	−1.4
*CASP6*	CASP6_BOVIN	Caspase-6	−1.5
*RPN2*	RPN2_BOVIN	Dolichyl-diphosphooligosaccharide-protein glycosyltransferase subunit 2	−1.5
*NIPSNAP3A*	G3X6L8_BOVIN	Uncharacterized protein	−1.6
*EIF3H*	EIF3H_BOVIN	Eukaryotic translation initiation factor 3 subunit H	−1.7
*YBX1*	YBOX1_BOVIN	Nuclease-sensitive element-binding protein 1	−1.7
*COPS7A*	F6QE33_BOVIN	Uncharacterized protein	−1.8
*PDLIM5*	G3MY19_BOVIN	Uncharacterized protein	−1.8
*MYBBP1A*	E1BKX3_BOVIN	Uncharacterized protein	−1.9
*RPL10*	RL10_BOVIN	60S ribosomal protein L10	−1.9
*EEA1*	F1MN61_BOVIN	Uncharacterized protein (Fragment)	−2.0
*ARPC1B*	ARC1B_BOVIN	Actin-related protein 2/3 complex subunit 1B	−2.2
*C789567*	A6H7H3_BOVIN	LOC789567	−2.5

**Table 3 ijms-20-01156-t003:** Primer sequences and annealing temperatures used in real-time reverse transcription-PCR assays as the size of the amplicons (in bp).

Gene Symbol	Primers Pair	Amplicon Size (bp)	T °C annealing	Reference
CCL5	AGC AGT TGT CTT TAT CAC CAG GA	87	60	[65]
	TCC AAA GCG TTG ATG TAC TCT C			
CD 36	ACA GAT GTG GCT TGA GCG TG	186	58	[63]
	ACT GGG TCT GTG TTT TGC AGG			
CSN2	CTC AAA CCC CTG TGG TGG TG	332	60	[1]
	AAA GGC CTG GAT GGG CAT AT			
IL1	GAA TGG AAA CCC TCT CTC CC	104	62	this article
	GCT GCA GCT ACA TTC TTC CC			
IL8	TGG GCC ACA CTG TGA AAA T	138	62	[66]
	TCA TGG ATC TTG CTT CTC AGC			
LAP	GAA ATT CTC AAA GCT GCC GTA	114	60	[67]
	TCC TCC TGC AGC ATT TTA CTT			
RBP4	CAA CGG TTA CTG TGA TGG	98	60	this article
	GAG GCT GAG TAA GGT TAA TG			
STAT3	GTC TAA CAA TGG CAG CCT CTC AGC	405	60	[68]
	AAG AGT TTC TCC GCC AGC GTC			
TAP	GCC AGC ATG AGG CTC CAT	166	60	[29]
	AAC AGG TGC CAA TCT GT			
INSIG1	CTA GCC TCG AAC TAA AGC CTG ACT	101	59	[69]
	TTC CTG TCT CAC CAC ACT TCA TCT			
**Housekeeping genes**				
UXT	TGT GGC CCT TGG ATA TGG TT	101	60	[63]
	GGT TGT CGC TGA GCT CTG TG			
EIF3K	CCA GGC CCA CCA AGA AGA A	125	60	[63]
	TTA TAC CTT CCA GGA GGT CCA TGT			
CLN3	TTC TGA CTC CTT GGG ACA CA	62	62	[63]
	CAA CCT GCC CAC CTA TCA GT			

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
