# Peer review of "Mammary Gland Transcriptome and Proteome Modifications by Nutrient Restriction in Early Lactation Holstein Cows Challenged with Intra-Mammary Lipopolysaccharide"

_ijms, 2019, doi:10.3390/ijms20051156_

Round 1

Reviewer 1 Report

The authors addressed well now to this reviewer's comments.

Author Response

We thank the reviewer for helping us to improve our manuscript in previous revisions and to allow our works to be publish.

Reviewer 2 Report

The authors made an attempt to address some main concerns I found in previous reviews. However, those related to the experimental design can not be changed, and they compromise the conclusions to be drawn from the study. In particular:

the inexistence of a pre-challenge sample. I understand that authors were affraid to biase their results due to any inflammatory reaction associated to the biopsy collection, even if it occurred at a different quarter. I think that it would not be a problem considering the asseptic collection and the short period until collection of the second sample.  Furthermore, considering the individual is the same, albeit some authors would consider that one quarter is not identical to the contralateral, but the response to feed restriction (the aim of this study) would be, expectabily/hopefully, similar.

Usually, we use microarrays to get a larger picture of possible differences in pathways; we then select one or two to be validated/studied in detail by  RT-PCR and  proteomics (to ascertain for post-transcriptional regulations of particular gene downstream products. However, in this study, the tools and targets seems to be selectedin and inverse or independent order.

the authors intend to force the message "feed resctriction = default mammary immune response". Despite this relation may exists, data obtained from this study do not support such hypothesis. The authors used an accute, short-term food restriction (4 days, 48% DMI) and tested the mammary gland  ability  to respond on inocultions performed on day 3 of the treatment. According to previous studies, I would expect the mammary gland to be coping with the altered body energy balance (with increse apoptosis and remodelation, along with changes in lipid and  protein precursors for milk. I think that data from microarrays and proteomic studies show it clearly. However, in contrast to the interprettion of the uthors, little can be extrapolate on regards to the immune status/competency of the mammary tissue. Authors mention the existence of data respecting some cytokines in milk, but choose to present data in a different paper.  Unfortunatly, that option reduces the soundeness of the present work

Most of the discussion content is bised by the authors biased speculation. Most of the ssumptions are not directly supported by the results of the current study.

the supplementary file needs to be revised 

I would like to send the authors a commented copy of the manuscript, to help them in future submissions.

Author Response

Response: Thank the reviewer for reviewing our manuscript. Thanks to such exchanges of the ideas which help us to improve it. Our experimental design was chosen for several reason. Indeed, it was observed that cows in the early lactation state are more susceptible to inflammation. The literature reported that NEB which occurs at that moment can play a role in this susceptibility. However, because of the complexity of the phenomena, NEB role in inflammation is still unclear. Our aim was to provide more new information which could help to understand this process. Our hypothesis was based on the information from literature and according to them, we choose our experimental design as explained in our previous response. We used transcriptomic and proteomic approaches to have an overview of the “NEB regulation” of pathways. It was still a global screening to give an exhaustive view of the modified processes as it is a case in all papers examine general gene expression with the microarray. Due to the inflammation status, and because it seems to us that the inflammatory aspect are important, we studied more specifically the know genes involved in the process by RT-qPCR before to use the global analyses. The global approaches pointed out genes involved in inflammatory process but there are few information about the exact role in the immune response. Their function is still not fully understood and describe even if they were found to play a role in. That is why we used the hypothesis as a compilation of our results and literature. We added figures to help to understand the complicity of the processes. These figures represent the role of our detected genes based on the knowledge from the literature which is the common way of investigation. All together, these knowledge is important to help prevent and fight against MG inflammation especially when nowadays it is big pressure to diminish the use of antibiotics in farm animals. The better understanding of these processes will help to breed animals and make genetic selection forward more resistant against inflammation.

We agreed that all data together would be complex but we decided to divide data in two parts: one dedicated to genomic analyses and another reported “systemic response” of the animal. I was impossible to put all data in the same paper. However we gave (and added in the revised version) information needed to interpret genomic results. We use our other paper as a reference (in JDS), the DOI of which we are waiting and will be published very soon. In addition, we would like also to point out that we have already included transcriptomic and proteomic analyzes in a same article, which is not often the case in other publications.

We would like to thank for the pdf file, we could follow your remarks and improve our paper. We have carefully considered all your comments and reply to each comment in attached pdf file.

Reviewer 3 Report

The authors have studied the effect of malnutrition on the inflammatory response of mammary glands in lactating cows. Transcriptome and proteome analysis have been used to study this effect. The authors conclude that malnutrition induced metabolic imbalance which inturn may affect the response to inflammation.

General comment

1.     Any reason why the authors would choose “undernutrition” for “malnutrition” in the transcript ?

2.     The font has to be made uniform throught the manuscript

3.     Color highlighting from reviews I guess still remains in the paper.

Minor changes

Abstract:

Line 26: The line here say one healthy rear mammary quarter misleads to say that only one animal was used for the transcriptome and proteome study. So maybe the sentence needs to rewrtitten as “one health rear mammary quarter from 12 lactating cows”.

Line 27: The line starting with as expected reports a confirmation bias in the analysis. Like the data has been used to prove what was expected. If this was not the case then the line needs to rewritten.

Line 50: Needs introduction to NEB.

Introduction

Line 41: “This pathology…” could be rewritten as ‘Its one of the most prevalent disease and has huge economic impact due to increased…. .

Line 63: Needs to grammatically corrected.

Line 63-68: Briefly mention what were the main conclusions of these studies.

Results:

Please add a paragraph about qualitative and quantitative description on selection of animals. This has been explained in the methods section, it in needs to be moved to the  results section.

Line 94: “was greater REST” should be change to “was greater in REST”

Table 1: Can be sorted based on fold change rather than alphabetically be gene, and maybe can be left centered.

Table 2: Can also be sorted based on fold change column.

Line 183: “which are involved in”

Author Response

Open Review (x) I would not like to sign my review report

( ) I would like to sign my review report

English language and style ( ) Extensive editing of English language and style required

(x) Moderate English changes required

( ) English language and style are fine/minor spell check required

( ) I don't feel qualified to judge about the English language and style

Yes         Can be improved             Must be improved          Not applicable

Does the introduction provide sufficient background and include all relevant references?             ( )            ( )            (x)                ( )

Is the research design appropriate?        (x)          ( )            ( )            ( )

Are the methods adequately described?              (x)          ( )            ( )            ( )

Are the results clearly presented?           ( )            (x)          ( )            ( )

Are the conclusions supported by the results?   (x)          ( )            ( )            ( )

Comments and Suggestions for Authors

The authors have studied the effect of malnutrition on the inflammatory response of mammary glands in lactating cows. Transcriptome and proteome analysis have been used to study this effect. The authors conclude that malnutrition induced metabolic imbalance which inturn may affect the response to inflammation.

General comment

1.     Any reason why the authors would choose “undernutrition” for “malnutrition” in the transcript ?

Response : We choose “undernutrition” as more adequate term in our opinion because it reflects the outcome of insufficient food intake. The term “malnutrition” results from eating a diet in which one or more nutrients are either not enough or are too much such that the diet causes health problems so to stress the fact that animals received less energetic food we decided to use the first term.

2.     The font has to be made uniform throught the manuscript

Response 2: Thanks to the reviewer to detect this mistake in the reference number and the tables.

3.     Color highlighting from reviews I guess still remains in the paper.

Response 3: Color highlighting does not appear in the version downloaded from the site. Probably, the editor modified it.

Minor changes

Abstract:

Line 26: The line here say one healthy rear mammary quarter misleads to say that only one animal was used for the transcriptome and proteome study. So maybe the sentence needs to rewrtitten as “one health rear mammary quarter from 12 lactating cows”.

Response: Changed

Line 27: The line starting with as expected reports a confirmation bias in the analysis. Like the data has been used to prove what was expected. If this was not the case then the line needs to rewritten.

Response: Changed

Line 50: Needs introduction to NEB.

Response: We thought that it was line 27 and changed the sentence

Introduction

Line 41: “This pathology…” could be rewritten as ‘Its one of the most prevalent disease and has huge economic impact due to increased…. .

Response: Changed

Line 63: Needs to grammatically corrected.

Response: Changed

Line 63-68: Briefly mention what were the main conclusions of these studies.

Response: Added

 Results:

Please add a paragraph about qualitative and quantitative description on selection of animals. This has been explained in the methods section, it in needs to be moved to the  results section.

Response: Thanks to the reviewer for this comment. We put those information in the methods section because it was a base for qualification of animals to our experiment so we treat it like part of used method. Moreover we were asked to put it in this part previously by other reviewer.

Line 94: “was greater REST” should be change to “was greater in REST”

Response: Done

Table 1: Can be sorted based on fold change rather than alphabetically be gene, and maybe can be left centered.

Response: We changed as requested. However to avoid excessive tracked modifications, we do not tracked these changes

Table 2: Can also be sorted based on fold change column.

Response: We changed as requested. However to avoid excessive tracked modifications, we do not tracked these changes

Line 183: “which are involved in”

Response: Done

Reviewer 4 Report

In the manuscript ‘Mammary Gland Transcriptome and Proteome Modifications by Nutrient Restriction in Early Lactation Holstein Cows Challenged with Intramammary Lipopolysaccharide” the authors aimed to test the hypothesis whether „undernutrition in early lactating cows would modify the inflammatory response at mRNA and protein levels”. It is not clear what is the practical meaning of this study. The aim of the study seems also to be not fully supported by the observation that undernutrition generally modifies the inflammatory response in the early lactation since, as the authors mention, this is also hypothetical (line 50: „…are likely to influence immune function”, line 56: „…that might influence immune system function”). Overall I have the impression that these are the secondary results of another study. The manuscript requires revision in some aspects as depicted below.

1.       Study design – why qPCR was conducted before microarray analysis. Ususally it is the other way round – qPCR is conducted to verify most significant changes observed in microarray study.

2.       Study design – there is some inconsistency between the aim of the study, the hypothesis and the methodology. The aim was focused specifically on the immune function but the „omics” analysis yielded data on many different processes. In fact, the process which was mostly affected by undernutrition at the transcriptome level, was associated with metabolism and the authors paid much attention to describe changes in metabolic processes induced by undernutrition after LPS challenge. Similarly, at the proteome level, the mainly affected process was not associated with immune response, but with protein synthesis and much of the discussion referes to this observation. From this point of view, the main aim of this study should be to investigate which processes are generally affected by undernutrition after LPS challenge.

3.       As already mentioned, what is practical meaning of this study? How may the industry benefit from this study results?

Minor comments

1.       Figures 1 and 2 – percent values would be helpful.  

Author Response

Open Review (x) I would not like to sign my review report

( ) I would like to sign my review report

English language and style ( ) Extensive editing of English language and style required

( ) Moderate English changes required

(x) English language and style are fine/minor spell check required

( ) I don't feel qualified to judge about the English language and style

Yes           Can be improved     Must be improved  Not applicable

Does the introduction provide sufficient background and include all relevant references?                             ( )            (x)                ( )            ( )

Is the research design appropriate?        ( )            (x)          ( )            ( )

Are the methods adequately described?              ( )            (x)          ( )            ( )

Are the results clearly presented?           ( )            (x)          ( )            ( )

Are the conclusions supported by the results?   ( )            (x)          ( )            ( )

Comments and Suggestions for Authors

In the manuscript ‘Mammary Gland Transcriptome and Proteome Modifications by Nutrient Restriction in Early Lactation Holstein Cows Challenged with Intramammary Lipopolysaccharide” the authors aimed to test the hypothesis whether „undernutrition in early lactating cows would modify the inflammatory response at mRNA and protein levels”. It is not clear what is the practical meaning of this study. The aim of the study seems also to be not fully supported by the observation that undernutrition generally modifies the inflammatory response in the early lactation since, as the authors mention, this is also hypothetical (line 50: „…are likely to influence immune function”, line 56: „…that might influence immune system function”). Overall I have the impression that these are the secondary results of another study. The manuscript requires revision in some aspects as depicted below.

Response: Thank to the reviewer for the commentary. It was observed that cows in the early lactation state are more susceptible to inflammation. The literature reported that NEB which occurs in that moment can play a role in this susceptibility. However, because of the complexity of the phenomena, NEB role in inflammation is still unclear. Our aim was to provide more new information which could help to understand this process. These knowledges are important to help to prevent and fight against MG inflammation especially when nowadays it is big pressure to diminish the use of antibiotics in farm animals. The better understanding of this process can help to breed animals and make genetic selection forward more resistant against inflammation. We used hypothetical statements because gene expression detected as modified still poorly documented. The hypothesis in this work was based on the literature and we focused on the role of DEG in inflammation. Indeed, our knowledge of function of all gene is still incomplete and sometimes there few information about them but its role can be important just still not fully understood.

1.       Study design – why qPCR was conducted before microarray analysis. Ususally it is the other way round – qPCR is conducted to verify most significant changes observed in microarray study.

Response: We clarified this point at the end of the introduction to explain our approaches, the title of the 2.2 section, and in the 3.1 section.

2.                   Study design – there is some inconsistency between the aim of the study, the hypothesis and the methodology. The aim was focused specifically on the immune function but the „omics” analysis yielded data on many different processes. In fact, the process which was mostly affected by undernutrition at the transcriptome level, was associated with metabolism and the authors paid much attention to describe changes in metabolic processes induced by undernutrition after LPS challenge. Similarly, at the proteome level, the mainly affected process was not associated with immune response, but with protein synthesis and much of the discussion referes to this observation. From this point of view, the main aim of this study should be to investigate which processes are generally affected by undernutrition after LPS challenge.

Response: We agree with the reviewer. We thought to have already clarified this point (with the last sentence of the introduction of the previous revision). However it appears to be not enough clear. So we added a sentence at the end of the introduction to clarify this point.  

3.                   As already mentioned, what is practical meaning of this study? How may the industry benefit from this study results?

Response: This study was performed with the aim to increase our knowledge on the effects of NEB occurring in early lactation which is the time of high risk of mastitis. Tis knowledge will be helpful for genetics studies. We added this point at the end of our manuscript.

Minor comments

1.       Figures 1 and 2 – percent values would be helpful.

Response: We added them.

Round 2

Reviewer 4 Report

I am satisfied with the improved version of the manuscript. I would only suggest to mention in the introduction at least one sentence about practical meaning. You explained it in the response to the reviewers but it is still missing in the text.

Author Response

We thank the reviewer for this last suggestion and added the sentences at the end of the introduction